# *N*-(Hydroxyalkyl) Derivatives of *tris*(1*H*-indol-3-yl)methylium Salts as Promising Antibacterial Agents: Synthesis and Biological Evaluation

**DOI:** 10.3390/ph13120469

**Published:** 2020-12-16

**Authors:** Sergey N. Lavrenov, Elena B. Isakova, Alexey A. Panov, Alexander Y. Simonov, Viktor V. Tatarskiy, Alexey S. Trenin

**Affiliations:** 1Gause Institute of New Antibiotics, 11 B. Pirogovskaya Street, 119021 Moscow, Russia; ebisakova@yandex.ru (E.B.I.); 7745243@mail.ru (A.A.P.); simonov-live@inbox.ru (A.Y.S.); as-trenin@mail.ru (A.S.T.); 2N. N. Blokhin Russian Cancer Research Center, 24 Kashirskoe Shosse, 115478 Moscow, Russia; tatarskii@gmail.com; 3National University of Science and Technology MISiS, 4 Leninsky Ave., 119049 Moscow, Russia

**Keywords:** *tris*(1*H*-indol-3-yl)methylium, turbomycin, indole derivatives, antibacterial action, overcoming of drug resistance

## Abstract

The wide spread of pathogens resistance requires the development of new antimicrobial agents capable of overcoming drug resistance. The main objective of the study is to elucidate the effect of substitutions in *tris*(1*H*-indol-3-yl)methylium derivatives on their antibacterial activity and toxicity to human cells. A series of new compounds were synthesized and tested. Their antibacterial activity in vitro was performed on 12 bacterial strains, including drug resistant strains, that were clinical isolates or collection strains. The cytotoxic effect of the compounds was determined using an test with HPF-hTERT (human postnatal fibroblasts, immortalized with hTERT) cells. The activity of the obtained compounds depended on the carbon chain length. Derivatives with C5–C6 chains were more active. The minimum inhibitory concentration (MIC) of the most active compound on Gram-positive bacteria, including MRSA, was 0.5 μg/mL. Compounds with C5–C6 chains also revealed high activity against *Staphylococcus epidermidis* (1.0 and 0.5 μg/mL, respectively) and moderate activity against Gram-negative bacteria *Escherichia coli* (8 μg/mL) and *Klebsiella pneumonia* (2 and 8 μg/mL, respectively). However, they have no activity against *Salmonella cholerasuis* and *Pseudomonas aeruginosa*. The most active compounds revealed higher antibacterial activity on MRSA than the reference drug levofloxacin, and their ratio between antibacterial and cytotoxic activity exceeded 10 times. The data obtained provide a basis for further study of this promising group of substances.

## 1. Introduction

In recent years, the situation in the field of therapy of infectious diseases has been significantly complicated due to the wide spread of pathogens resistant to known antibiotic drugs [1,2,3]. To solve this problem, it is proposed to try and prevent the drug resistance by rational use of antibiotics, and by development of new antimicrobial agents capable of overcoming drug resistance. The latter direction is one of the most important ones of modern medicinal chemistry [4]. The most promising compounds are the ones that have low toxicity for human cells and retain high activity on resistant strains of pathogens. The main objective of this study was to elucidate the effect of substituents on the activity of compounds that contain *tris*(1-alkylindol-3-yl)methylium core with respect to various test microorganisms, as well as their toxicity for human cells.

Recently, compounds containing triphenylmethyl or triindolylmethyl fragments have attracted the interest of researchers. This is due to the fact that some compounds of this type exhibit useful biological properties, such as antimicrobial and antiproliferative [5,6,7,8,9,10,11,12,13].

Earlier, we studied the structure-antimicrobial and cytotoxic activity relationship among a new class of compounds—*tris*(1-alkylindol-3-yl)methylium salts, structurally similar to the natural antibiotic turbomycin A [5]. Among them, a number of substances with a high (submicromolar) activity, even on multidrug-resistant strains of *Staphylococcus aureus*, were found. The first symmetrical alkyl derivatives we obtained were highly active against bacteria, but their toxicity was higher than the antibacterial activity [11,12]. Their structure needed to be improved. One of the successful attempts were chimeric structures [14] combining fragments of *tris*(1-alkylindol-3-yl)methylium and 3,4-disubstituted pyrrole-2,5-diones (maleinimides) [15,16]. They had a relatively low toxicity to human cells, and at the same time, a high activity on resistant strains of Gram-positive bacteria [14], acting by disrupting the functioning of their membranes [17]. 

When analyzing the structure-activity relationship of these compounds, we observed that the activity of these substances is closely related to their lipophilicity. The most promising substances had a LogP_ow_ (partition between octanol and water) in the region of 2–5, and if the LogP_ow_ was lower than that, the substance had neither pronounced antibacterial activity nor cytotoxicity. If the LogP_ow_ was higher then 5, the cytotoxicity was higher than the antibacterial activity. Thus, the assumption was made about the optimal region of LogPow, where one can expect to find substances with a good ratio of antibacterial activity to cytotoxicity. Symmetrical *N*-(hydroxyalkyl) derivatives of *tris*(1*H*-indol-3-yl)methylium seem very promising in this regard, since by changing the length of the hydrocarbon part of the substituent, it is possible to adjust the lipophilicity of the molecule. In addition, the presence of hydroxyl groups improves the solubility of substances in aqueous media. This paper presents the synthesis and study of antibacterial and cytotoxic activity in the homologous series of *N-*(hydroxyalkyl) derivatives of *tris*(1*H*-indol-3-yl)methylium with a hydrocarbon chain length from C2 to C6.

## 2. Results and Discussion

### 2.1. Chemistry

The title compounds were synthesized according to Scheme 1. To start, reagents 2-(1*H*-indol-1-yl)acetic acid (**1a**) and 3-(1*H*-indol-1-yl)propanoic acid (**1b**) were used, which were converted to the corresponding (1*H*-indol-1-yl)alkanols **3a** and **3b** by LiAlH_4_ reduction. To obtain (1*H*-indol-1-yl)alkanols **3c–e**, another synthesis method was used—alkylation of indole by ω-bromoalkanols **2c–e**. The hydroxyl group of compounds **3a–e** was then protected by acetylation to obtain substances **4a–e**, which were then subjected to formylation by the Wilsmeier–Haack method, with further deacetylation to obtain 1-(ω-hydroxyalkyl)-1*H*-indole-3-carbaldehydes **6a–e**. Without the protection of the hydroxyl group, formylation of **3a–e** is not possible, the reaction leading to formation of a complex mixture of products. To obtain symmetric *tris*(1-[ω-hydroxyalkyl]-1*H*-indol-3-yl)methanes **7a–e**, we condensed (1*H*-indol-1-yl)alkanoles **3c–e** with the corresponding aldehydes **6a–e** in 2:1 molar ratio in boiling methanol using dysprosium triflate Dy(OTf)_3_ as a catalyst. As a final stage, compounds **7a–e** were oxidized by DDQ (2,3-dichloro-5,6-dicyano-1,4-benzoquinone) in THF with subsequent HCl treatment to yield the target *tris*(1-[ω-hydroxyalkyl]-1H-indol-3-yl)methylium salts **8a–e.** All synthesized compounds were characterized by NMR spectroscopy, high-resolution mass spectrometry (HRMS), and HPLC.

### 2.2. Biological Evaluation

#### 2.2.1. Antibacterial Activity

As reference compounds, we used levofloxacin, a common broad-spectrum antibiotic, and Brilliant Green, a common antiseptic with a structure somewhat similar to substances **8a–e,** being triarylmethylium salt. The compounds **7a–e** were insoluble in water, so for a test for antibacterial activity, a solubilizer had to be used. Addition of Kolliphor EL (5× by weight) provided enough solubility for the compounds. The compounds **8a–e** were soluble enough in water by themselves.

Trisidolylmethanes **7a–e** showed practically no antibacterial activity (MICs >64 μg/mL). This is in good agreement with our earlier data that triindolylmethanes are biologically inactive until they are oxidized to trisindolylmethylium salts [11,12,14]. Trisindolylmethylium salts **8a–e** exhibited significant antibacterial activity, as shown in Table 1. The data obtained show that **8a–b** substituted with C2 and C3 hydroxyalkyl substituents have practically no antibacterial activity, however, with further growth of the chain length starting from C4, pronounced activity appears, reaching 0.5 µg/mL in the C6 derivative (compound **8e**). The compounds which are active on bacteria but are low-toxic should balance between pore-forming activity on the lipid bilayers and non-selective detergent activity [17,18].

Compounds **8d** and **8e** were highly active against Gram-positive bacteria, including strains with antibiotic resistance. For example, activity of compound **8e** against *S. aureus* ATCC 25,923 and clinical isolate *S. aureus* 10, that were sensitive to all antibiotics, was 0.5 μg/mL. At the same time, activity of this compound against two methicillin resistant strains (MRSA) (*S. aureus* 5 and *S. aureus* 100 KC) was just the same high (0.5 μg/mL). The same activity (0.5 μg/mL) was revealed against *S. aureus* ATCC 3798, which was resistant not only to ampicillin, oxacillin, cefuroxime, and carbenicillin (antibiotics of penicillin and cefalosporine group), but also to clindamycin, erythromycin, rifampicin, ciprofloxacin, and levofloxacin. Compounds **8d** and **8e** were active against *S. aureus* ATCC 700699, which possess resistance to levofloxacin. 

Compounds **8d** and **8e** were also active (1, 0.5 μg/mL) against *Staphylococcus epidermidis* 533, which is resistant to gentamicin, but, in contrast to **8d** (8 μg/mL), compound **8e** was almost inactive (>64 μg/mL) to *Enterococcus faecium* 569, which possess resistance to cefuroxime, clindamycin, gentamycin, vancomycin, and doxycycline. 

A moderate level of activity of **8d** and **8e** was found against *Escherichia coli* ATCC 25,922 and *Klebsiella pneumoniae* ATCC 13883. Against *K. pneumoniae* compound **8d** was slightly more active (2 μg/mL) than compound **8e** (8 μg/mL). However, it should be noted that these two compounds, in general, were significantly less active against Gram-negative bacteria than against Gram-positive ones. 

#### 2.2.2. Cytotoxic Activity

Compounds **7a–e** all showed similar cytotoxicity with IC_50_ higher than 50 μg/mL. Cytotoxicity of compounds **8a–e** (Table 1) depended on the length of the hydroxyalkyl chain and for C2–C4 substituents IC_50_, were higher than 50 µg/mL. For C5 compound **8d**, it was 13 µg/mL, and then in C6, there is a sharp increase in cytotoxicity, which reached 2 µg/mL. Apparently, this rapid increase in cytotoxicity is associated with an increase in the total detergent activity of the molecule, which leads to a decrease in the selectivity of the action on the lipid layers of the membrane [17,18].

### 2.3. Study of the Relationship between Lipophility and Biological Activity

Analyzing the structure-activity relationship for compounds **8a–e**, it was observed that the activity of these substances is closely related to their lipophilicity. There are two possible tautomeric variations of methylium (Figure 1): form 1, where the positive charge is on the central carbon atom, and closer to the real form 2, where the positive charge is on one of the nitrogen atoms of the indole cycles.

It was shown that the calculated values in this case are very close to the real ones when the contribution of both forms is taken into account. In other words, the arithmetic mean (miLogP_ow form1_ + miLogP_ow form2_)/2 will be closer to the real LogP_ow_ obtained experimentally (Table 2). Data analysis shows that molecules with a good ratio of antibacterial activity to cytotoxicity are more likely to be in the LogP range from 2 to 5. Above 5, a sharp increase in cytotoxicity begins, and below 2, there is an almost complete absence of antimicrobial activity. The best result is likely to be expected with a LogP in the region of 4. Computer calculations of lipophilicity in the Molinspiration package fairly adequately predict LogP for molecules and can be used to select potentially promising compounds of this group.

## 3. Materials and Methods

### 3.1. Chemistry

All the reagents were obtained commercially and used without further purification. Indole and all ω-bromoalkanols, all solvents, LiAlH_4_, Dy(OTf)_3_, DMAP, DDQ were purchased from Sigma-Aldrich; 2-(1*H*-indol-1-yl)acetic acid and 3-(1*H*-indol-1-yl)propanoic acid were purchased from Alinda company (www.alinda.ru). Purity of the compounds was checked by thin layer chromatography using silica-gel 60 F_254_-coated Al plates (Merck) and spots were observed under UV light (254 nm). Column chromatography was performed on Kieselgel 60 (Merck). Proton nuclear magnetic resonance (^1^H NMR) and carbon-13 nuclear magnetic resonance (^13^C NMR) spectra (in DMSO-d6) were recorded on a Varian VXR-400 spectrometer at 400 and 100 MHz respectively, the chemical shift values are expressed in ppm (δ scale) using DMSO as an internal standard, the coupling constants expressed in Hz. The NMR spectra of the compounds **8a–e** were recorded at 80 °C to avoid peaks broadening. The mass spectral measurements were carried out by ESI method on microTOF-QII (Brucker Daltonics GmbH). Analytical high-performance liquid chromatography (HPLC) was performed on a Shimadzu LC-20AD system using Kromasil-100-5-C18 (Akzo-Nobel) column, 4.6 × 250 mm, 20 °C temperature, UV detection, mobile phase A—0.2% HCOONH_4_), mobile phase B-MeCN, (pH 7.4), fl-1 mL/min., loop 20 mkl. The NMR spectra of the compounds **3–10** are presented in Appendix A.

### 3.2. Antibacterial Activity

Compounds were tested against Gram-positive and Gram-negative bacteria, including sensitive or drug resistant strains from American Type Culture Collection (ATCC), as well as resistant clinical isolates from the culture collection of the Laboratory for Control of Hospital Infections (Sechenov University, Moscow, Russia). Collection cultures of Gram-positive bacteria: *Staphylococcus aureus* ATCC 25923, *Staphylococcus aureus* ATCC 3798, *Staphylococcus aureus* ATCC 700,699 and clinical isolates of Gram-positive bacteria: *Staphylococcus aureus* 5, *Staphylococcus aureus* 10, *Staphylococcus aureus* 100 KS, *Staphylococcus epidermidis* 533, *Enterococcus faecium* 569 and collection cultures of Gram-negative bacteria: *Escherichia coli* ATCC 25922, *Klebsiella pneumoniae* ATCC 13883, *Salmonella cholerasuis* ATCC 14,028 were used.

For the cultivation of the strains, various nutrient media were used: Trypticase Soy Agar BBL for *Staphylococcus* sp., *E. coli*, *K. pneumoniae*, *S. cholerasuis* and Columbia Agar Base BBL for the cultivation of *Enterococcus* sp. Cultures grown on appropriate nutrient media at 35 °C for 1 day were used to set up experiments. For determination of the antibacterial action Mueller-Hinton (Acumedia, Baltimore, MD, USA) liquid medium was used. The minimum inhibitory concentrations (MIC) were determined by the microdilution method in 96 well sterile plates in a cation-adjusted Müller-Hinton medium in accordance with the requirements of the Institute of Clinical and Laboratory Standards (CLSI/NCCLS) [19]. MIC was defined as the minimum drug concentration that completely prevents the growth of the test organism.

### 3.3. Cytotoxic Activity

The cytotoxic properties of the compounds obtained were tested using the MTT assay as described previously [11] on the healthy donor (postnatal) human fibroblasts immortalized by transfection of the hTERT gene of the catalytic component of telomerase (hereinafter, FB).

Cells were grown at 37 °C and 5% CO_2_. Human donor fibroblasts were cultivated in DMEM medium (Paneko, Russia) with addition of 10% FBS (Hyclone, Austria), 2 mM L-glutamine (Paneko, Russia), and 1% penicillin-streptomycin (Paneko, Russia). Cells were seeded at concentration 2500 cells/well in 96 well plates (Corning, NY, USA), and left overnight to attach. The next day, the cells were treated with compounds, with indicated concentrations (ten two-fold dilutions, starting from 50 uM) for 72 h. After incubation, MTT reagent (3-(4,5-dimethylthiazol-2-yl)-2,5-diphenyl tetrazolium bromide) (Sigma-Aldrich, Saint-Louis, MO, USA) was added to a final concentration of 0.5 ug/mL and the cells were incubated for 2 h at 37 °C and 5% CO_2_. After incubation, the medium was discarded, and 100 uL of DMSO was added. The optical densities were read at 570 nm wavelength on Multiskan FC (ThermoFisher, Waltham, MA, USA). The OD values for controls were taken as 100%. The IC_50_ values were calculated in GraphPadRrism 6.0.

### 3.4. Determination of Lipophilicity

We used the partition coefficient as an indicator of lipophilicity. The partition coefficient (P) is defined as the ratio of the equilibrium solute concentrations in a two-phase system of immiscible solvents. The most common in practice is the octanol-water (P_ow_) system. The partition coefficient is usually represented as a decimal logarithm (LogP_ow_). It can be measured in several ways. A most advanced, accurate, and less time-consuming is the HPLC method for determining P_ow_ using high-performance liquid chromatography [20]. HPLC is performed on an analytical column with a solid phase containing long hydrocarbon chains chemically bound to silica gel. The retention time on such a column (R_t_) is directly related to the partition coefficient P_ow_. The most informative in our case was the partition coefficient at a physiological pH value of 7.4. It was at this pH that the main biological experiments with the studied substances were carried out, namely tests of antimicrobial activity and cytotoxicity. HPLC was performed on a Shimadzu LC-20AD system using Kromasil-100-5-C18 (Akzo-Nobel) column, 4.6 × 250 mm, 20 °C temperature, UV detection, mobile phase A—0.2% HCOONH_4_), mobile phase B-MeCN, (pH 7.4), fl-1 mL/min., loop 20 mkl. After calibration using substances with a known LogP_ow_, it is possible to recalculate R_t_ to LogP_ow_. For calibration, we used aniline (LogP_ow_ 0.9), p-chloroaniline (LogP_ow_ 1.8), diphenylamine (LogP_ow_ 3.4), triphenylamine (LogP_ow_ 5.7).

To study the possibility of using computer models to calculate LogP_ow_ [21], miLogP_ow_ values were calculated for the same substances by the Molinspiration package [22], which is an online tool available at www.molinspiration.com. The method for logP prediction developed at Molinspiration (miLogP_ow_) is based on group contributions. These have been obtained by fitting calculated logP with experimental logP for a training set more than twelve thousand, mostly drug-like molecules. In this way, hydrophobicity values for 35 small simple “basic” fragments were obtained, as well as values for 185 larger fragments, characterizing intramolecular hydrogen bonding contribution to logP and charge interactions. Molinspiration methodology for logP calculation is very robust and is able to process practically all organic molecules. For 50.5% of molecules, logP is predicted with error <0.25, for 80.2% with error <0.5 and for 96.5% with error <1.0. Only for 3.5% of structures, logP is predicted with error >1.0. The statistical parameters listed above rank Molinspiration miLogP as one of the best methods available for logP prediction. MiLogP is used due to its robustness and good prediction quality in the popular ZINC database for virtual screening. A report by the National Institute of Standards documenting excellent agreement between experimental logP and Molinspiration calculated logP for some industrial chemicals [23].

### 3.5. Chemical Experimental Data

2-(1*H*-Indol-1-yl)-ethanol **(3a)**.

To the boiling suspension of LiAlH_4_ (15.2 g, 0.4 mol) in THF (500 mL), the solution of 2-(1*H*-indol-1-yl)acetic acid **1a** (17.56 g, 0.1 mol) in THF (100 mL) was gradually added, then the reaction mixture was refluxed for 5 h. After cooling to RT, the reaction mixture was quenched with KOH (20% aqueous solution), then was filtered and diluted with EtOAc (300 mL) and aqueous solution of citric acid (10.0 g in 100 mL) was added. The organic layer was separated, washed with water and brine, and evaporated in vacuo. The residue was purified by flash chromatography (50 g of silica gel) using EtOAc-hexane (1:10 to 1:1) as an eluent, to give **3a** (13.2 g, 82%) as a colorless oil.

^1^H NMR: δ 7.52 (dt, 1H, J = 7.8, 1.1 Hz), 7.48–7.41 (m, 1H), 7.33 (d, 1H, J = 3.1 Hz), 7.10 (ddd, 1H, J = 8.2, 7.0, 1.3 Hz), 6.99 (ddd, 1H, J = 8.0, 7.0, 1.0 Hz), 6.40 (dd, 1H, J = 3.1, 0.9 Hz), 4.93 (t, 1H, J = 5.3 Hz), 4.19 (t, 2H, J = 5.7 Hz), 3.70 (q, 2H, J = 5.5 Hz). ^13^C NMR: δ 136.33, 129.59, 128.53, 121.26, 120.73, 119.24, 110.32, 100.67, 60.76, 48.66, 48.64. HRMS (EI) m/z [M + H]^+^ Calcd for C_10_H_12_NO^+^ 162.0913; Found 162.0923.

3-(1*H*-Indol-1-yl)propan-1-ol **(3b)**.

The same procedure as above was carried out using 3-(1*H*-indol-1-yl)propanoic acid (18.9 g, 0.1 mol), to give **3b** (13.8 g, 79%) as a colorless oil.

^1^H NMR: δ 7.53 (dt, 1H, J = 7.8, 1.0 Hz), 7.44 (dd, 1H, J = 8.3, 1.0 Hz), 7.32 (d, 1H, J = 3.1 Hz), 7.14–7.07 (m, 1H), 7.04–6.97 (m, 1H), 6.41 (dd, 1H, J = 3.2, 0.9 Hz), 4.69 (s, 1H), 4.21 (t, 2H, J = 6.9 Hz), 3.37 (d, 2H, J = 5.2 Hz), 1.88 (t, 2H, J = 6.6 Hz). ^13^C NMR: δ 136.08, 129.12, 128.53, 121.38, 120.85, 119.27, 110.17, 100.82, 58.27, 42.84, 33.43. HRMS (EI) m/z [M + H]^+^ Calcd for C_11_H_14_NO^+^ 176.1070; Found 176.1073.

4-(1*H*-Indol-1-yl)butan-1-ol **(3c)**.

To the suspension of KOH (20 g, 0.35 mol) in DMSO (100 mL), indole (11.7 g, 0.1 mol) and 4-bromobutan-1-ol (16.8 g, 0.11 mol) were added. After intensive stirring at RT for 5 h, the reaction mixture was filtered, diluted with EtOAc (300 mL), and washed with an aqueous solution of citric acid (10.0 g, 100 mL). The organic layer was separated, washed with water and brine, and evaporated in vacuo. The residue was purified by flash chromatography (50 g of silica gel) using EtOAc-Hexane (1:10 to 1:1) as an eluent, to give **3c** (17.5 g, 93%) as a colorless oil.

^1^H NMR: δ 7.53 (dt, 1H, J = 7.8, 1.0 Hz), 7.44 (dd, 1H, J = 8.3, 1.1 Hz), 7.33 (d, 1H, J = 3.1 Hz), 7.11 (ddd, 1H, J = 8.2, 6.9, 1.2 Hz), 7.00 (ddd, 1H, J = 8.0, 7.0, 1.0 Hz), 6.41 (dd, 1H, J = 3.1, 0.9 Hz), 4.47 (t, 1H, J = 5.1 Hz), 4.15 (t, 2H, J = 7.1 Hz), 3.39 (td, 2H, J = 6.4, 4.9 Hz), 1.77 (dq, 2H, J = 9.6, 7.2 Hz), 1.43–1.32 (m, 2H). ^13^C NMR: δ 136.11, 129.00, 128.57, 121.34, 120.85, 119.23, 110.20, 100.79, 60.80, 45.84, 30.21, 27.09. HRMS (EI) m/z [M + H]^+^ Calcd for C_12_H_18_NO^+^ 190.1226; Found 190.1228.

5-(1*H*-Indol-1-yl)pentan-1-ol **(3d)**.

The same procedure as above was carried out using indole (11.7 g, 0.1 mol) and 5-bromopentan-1-ol (18.3 g, 0.11 mol), to give **3d** (18.1 g, 89%) as a colorless oil.

^1^H NMR: δ 7.54 (dt, 1H, J = 7.9, 1.0 Hz), 7.42 (dd, 1H, J = 8.2, 1.1 Hz), 7.31 (d, 1H, J = 3.2 Hz), 7.12 (ddd, 1H, J = 8.2, 7.0, 1.3 Hz), 7.01 (ddd, 1H, J = 8.0, 6.9, 1.0 Hz), 6.41 (dd, 1H, J = 3.1, 0.9 Hz), 4.43 (s, 1H), 4.11 (t, 2H, J = 7.0 Hz), 3.37 (t, 2H, J = 6.5 Hz), 1.73 (p, 2H, J = 7.2 Hz), 1.49–1.37 (m, 2H), 1.33–1.19 (m, 2H). ^13^C NMR: δ 136.10, 128.97, 128.57, 121.35, 120.86, 119.23, 110.14, 100.80, 61.06, 45.95, 32.54, 30.23, 23.37. HRMS (EI) m/z [M + H]^+^ Calcd for C_13_H_18_NO^+^ 204.1383; Found 204.1381.

6-(1*H*-Indol-1-yl)hexan-1-ol **(3e)**.

The same procedure as above was carried out using indole (11.7 g, 0.1 mol) and 6-bromohexan-1-ol (20.0 g, 0.11 mol), to give **3e** (19.5 g, 90%) as a colorless oil.

^1^H NMR: δ 7.22–7.15 (m, 1H), 7.09–6.98 (m, 3H), 3.52 (t, 1H, J = 6.8 Hz), 2.85 (t, 1H, J = 7.2 Hz), 2.20 (d, 3H, J = 11.3 Hz), 1.77 (p, 1H, J = 6.9 Hz). ^13^C NMR: δ 170.04, 165.67, 151.95, 136.37, 135.88, 134.80, 132.56, 130.15, 129.63, 125.82, 42.19, 37.21, 30.97, 28.14, 20.96, 20.84. HRMS (EI) m/z [M + H]^+^ Calcd for C_14_H_20_NO^+^ 218.1539; Found 218.1540.

2-(1*H*-Indol-1-yl)ethyl acetate **(4a)**.

To the solution of 2-indol-1-yl-ethanol **3a** (12 g, 75 mmol) in pyridine (100 mL), Ac_2_O (8 mL, 10 mmol) and DMAP (100 mg, 0.08 mmol) were added. After stirring at rt for 5 h, the reaction mixture was evaporated in vacuo, diluted with EtOAc (300 mL) and washed with an aqueous solution of citric acid (1.0 g in 100 mL). The organic layer was separated, washed with water and brine and evaporated in vacuo. The residue was purified by flash chromatography (100 g of silica gel) using EtOAc-Hexane (1:10 to 1:3) as an eluent, to give **4a** (14.7 g, 97%) as a colorless oil. ^1^H NMR: δ 7.63 (d, 1H, J = 7.8 Hz), 7.50 (dd, 1H, J = 8.3, 0.6 Hz), 7.35 (d, 1H, J = 3.2 Hz), 7.26–7.16 (m, 1H), 7.11 (td, 1H, J = 7.5, 0.9 Hz), 6.52 (dd, 1H, J = 3.1, 0.7 Hz), 4.37 (qd, 4H, J = 6.2, 1.5 Hz), 1.93 (s, 3H). ^13^C NMR: δ 170.53, 136.43, 129.16, 128.75, 121.63, 120.96, 119.58, 110.09, 101.50, 63.40, 44.93, 20.86.

3-(1*H*-Indol-1-yl)propyl acetate **(4b)**.

The same procedure as above was carried out using 3-indol-1-yl-propan-1-ol **3b** (7.0 g, 40 mmol), to give **4b** (8.1 g, 94%) as a colorless oil. ^1^H NMR: δ 7.62 (d, 1H, J = 7.8 Hz), 7.46 (d, 1H, J = 8.2 Hz), 7.31 (d, 1H, J = 3.1 Hz), 7.24–7.15 (m, 1H), 7.15–7.06 (m, 1H), 6.55 – 6.47 (m, 1H), 4.22 (t, 2H, J = 6.8 Hz), 3.96 (t, 2H, J = 6.4 Hz), 2.14–1.96 (m, 5H). ^13^C NMR: δ 170.75, 136.20, 128.81, 128.74, 121.54, 120.97, 119.43, 110.00, 101.23, 61.71, 42.73, 29.29, 20.92.

4-(1*H*-Indol-1-yl)butyl acetate **(4c)**.

The same procedure as above was carried out using 4-(1*H*-indol-1-yl)butan-1-ol **3c** (10.0 g, 53 mmol), to give **4c** (11.7 g, 96%) as a colorless oil. ^1^H NMR: δ 7.53 (d, 1H, J = 7.8 Hz), 7.47–7.39 (m, 1H), 7.32 (d, 1H, J = 3.1 Hz), 7.15–7.05 (m, 1H), 7.04–6.96 (m, 1H), 6.41 (dd, 1H, J = 3.1, 0.7 Hz), 4.15 (t, 2H, J = 7.0 Hz), 3.95 (t, 2H, J = 6.6 Hz), 1.94 (s, 3H), 1.92–1.69 (m, 2H), 1.69–1.42 (m, 2H). ^13^C NMR: δ 170.85, 136.06, 128.93, 128.55, 121.40, 120.86, 119.28, 110.12, 100.92, 100.89, 85.49, 63.84, 45.44, 26.84, 25.99, 21.06, 21.05. 

5-(1*H*-Indol-1-yl)pentyl acetate **(4d)**.

The same procedure as above was carried out using 5-(1*H*-indol-1-yl)pentan-1-ol **3d** (10.0 g, 49 mmol), to give **4d** (11.4 g, 95%) as a colorless oil. ^1^H NMR: δ 7.52 (d, J = 7.8 Hz, 2H), 7.47–7.38 (m, 2H), 7.32 (d, J = 3.1 Hz, 2H), 7.15–7.05 (m, 2H), 7.04–6.94 (m, 2H), 6.40 (dd, J = 3.0, 0.6 Hz, 2H), 4.12 (t, J = 7.0 Hz, 4H), 3.92 (t, J = 6.6 Hz, 4H), 2.48 (s, 1H), 1.94 (s, 6H), 1.82–1.64 (m, 4H), 1.62–1.47 (m, 4H), 1.23 (dd, J = 9.2, 6.2 Hz, 4H). ^13^C NMR: δ 170.84, 136.03, 128.97, 128.51, 121.33, 120.83, 119.22, 110.14, 100.77, 64.07, 45.69, 29.86, 28.10, 23.15, 21.10.

6-(1*H*-Indol-1-yl)hexyl acetate **(4e)**.

The same procedure as above was carried out using 6-(1*H*-indol-1-yl)hexan-1-ol **3e** (10.0 g, 46 mmol), to give **4e** (10.97 g, 93%) as a colorless amorphous solid. ^1^H NMR: δ 7.51 (d, 2H, J = 7.8 Hz), 7.42 (d, 2H, J = 8.2 Hz), 7.31 (d, 2H, J = 3.0 Hz), 7.09 (d, 2H, J = 7.3 Hz), 6.99 (d, 2H, J = 7.3 Hz), 6.39 (d, 2H, J = 2.7 Hz), 4.12 (t, 4H, J = 7.0 Hz), 3.92 (t, 4H, J = 6.6 Hz), 1.95 (s, 6H), 1.72 (d, 3H, J = 7.2 Hz), 1.69–1.25 (m, 10H), 1.25–0.98 (m, 5H). ^13^C NMR: δ 170.86, 136.05, 128.95, 128.51, 121.31, 120.82, 119.20, 110.11, 100.75, 64.16, 45.76, 30.15, 28.44, 26.34, 25.45, 21.11.

2-(3-Formyl-1*H*-indol-1-yl)ethyl acetate **(5a)**.

2-(1*H*-Indol-1-yl)ethyl acetate **4a** (14.7 g, 72 mmol) was dissolved in the solution of POCl_3_ (0.9 mL, 10 mmol) in DMF (50 mL) and intensively stirred at 5 °C for 5 h. The reaction mixture was quenched with Na_2_CO_3_ (10% aqueous solution), diluted with EtOAc (100 mL) and water (200 mL). The organic layer was separated and the water layer was re-extracted with EtOAc (100 mL). The combined extracts were washed with water and brine and evaporated in vacuo. The residue was purified by flash chromatography (100 g of silica gel) using EtOAc-Hexane (1:5 to 1:1) as an eluent to give **5a** (11.9 g, 71%) as a colorless oil. ^1^H NMR: δ 9.94 (s, 1H), 8.27 (s, 1H), 8.16 (d, 1H, J = 7.3 Hz), 7.61 (d, 1H, J = 7.9 Hz), 7.36–7.21 (m, 2H), 4.51 (t, 2H, J = 5.0 Hz), 4.38 (t, 2H, J = 5.1 Hz), 1.88 (s, 3H). ^13^C NMR: δ 185.18, 170.48, 141.46, 137.60, 125.09, 124.04, 122.98, 121.56, 117.98, 111.35, 62.73, 45.81, 20.84.

3-(3-Formyl-1*H*-indol-1-yl)propyl acetate **(5b)**.

The same procedure as above was carried out using 3-(1*H*-indol-1-yl)propylacetate **4b** (8.0 g, 36.8 mmol), to give **5b** (6.1 g, 68%) as a colorless amorphous solid. 1H NMR δ 9.91 (s, 1H), 8.27 (s, 1H), 8.19–8.11 (m, 1H), 7.58 (d, J = 8.0 Hz, 1H), 7.33–7.21 (m, 2H), 4.32 (t, J = 6.9 Hz, 2H), 3.96 (t, J = 6.2 Hz, 2H), 3.53 (s, 2H), 2.18–2.02 (m, 2H), 1.91 (s, 3H). ^13^C NMR: δ 184.98, 170.77, 170.76, 141.11, 137.43, 125.13, 124.00, 122.92, 121.56, 117.73, 111.29, 61.61, 43.80, 28.75, 20.91.

4-(3-Formyl-1*H*-indol-1-yl)butyl acetate **(5c)**.

The same procedure as above was carried out using 4-(1*H*-indol-1-yl)butyl acetate **4c** (11.7 g, 50 mmol), to give **5c** (9.7 g, 74%) as a colorless amorphous solid. ^1^H NMR: δ 9.89 (s, 1H), 8.30 (s, 1H), 8.10 (d, 1H, J = 7.5 Hz), 7.61 (d, 1H, J = 8.1 Hz), 7.33–7.20 (m, 2H), 4.28 (t, 2H, J = 7.1 Hz), 3.98 (t, 2H, J = 6.6 Hz), 1.94 (s, 3H), 1.91–1.75 (m, 2H), 1.75–1.47 (m, 2H). ^13^C NMR: δ 184.99, 170.84, 141.12, 137.41, 125.11, 123.97, 122.91, 121.51, 117.55, 111.46, 85.48, 63.71, 46.32, 26.39, 25.83, 21.08.

5-(3-Formyl-1*H*-indol-1-yl)pentyl acetate **(5d)**.

The same procedure as above was carried out using 5-(1*H*-indol-1-yl)pentyl acetate **4d** (11.4 g, 46 mmol), to give **5d** (9.78 g, 77%) as a colorless amorphous solid. ^1^H NMR: δ 9.90 (s, 1H), 8.30 (s, 1H), 8.12 (d, 1H, J = 7.4 Hz), 7.61 (d, 1H, J = 8.1 Hz), 7.34–7.22 (m, 2H), 4.26 (t, 2H, J = 7.1 Hz), 3.95 (t, 2H, J = 6.6 Hz), 1.94 (s, 3H), 1.88–1.73 (m, 2H), 1.64–1.51 (m, 2H), 1.34–1.06 (m, 2H). ^13^C NMR: δ 184.57, 170.46, 140.76, 137.06, 124.74, 123.57, 122.50, 121.13, 117.13, 111.09, 63.61, 46.18, 28.94, 27.63, 22.60, 20.70. 

6-(3-Formyl-1*H*-indol-1-yl)hexyl acetate **(5e)**.

The same procedure as above was carried out using 6-(1*H*-indol-1-yl)hexyl acetate **4e** (10.9 g, 42 mmol), to give **5e** (8.8 g, 73%) as a colorless amorphous solid. ^1^H NMR: δ 9.87 (s, 1H), 8.30 (s, 1H), 8.09 (d, 1H, J = 7.6 Hz), 7.61 (d, 1H, J = 8.2 Hz), 7.29 (dt, 2H, J = 8.2, 1.3 Hz), 7.23 (dt, 2H, J = 7.0, 1.0 Hz), 4.29 (t, 2H, J = 7.1 Hz), 3.92 (t, 2H, J = 7.0 Hz), 1.94 (s, 3H), 1.82–1.75 (m, 2H), 1.53–1.46 (m, 2H) 1.34–1.22 (m, 4H). ^13^C NMR: δ 184.89, 170.55, 140.86, 137.15, 124.88, 123.53, 122.59, 121.19, 117.20, 111.01, 64.14, 46.64, 29.59, 28.37, 26.14, 25.38, 21.13.

1-(2-Hydroxyethyl)-1*H*-indole-3-carbaldehyde **(6a)**.

2-(3-Formyl-1*H*-indol-1-yl)ethyl acetate **5a** (11.9 g, 51 mmol) was dissolved in the solution Na (200 mg, 0.8 mmol) in MeOH (50 mL) and stirred at 10 min. Then, the reaction mixture was evaporated in vacuo, quenched with an aqueous solution of citric acid (0.5 g, 50 mL), and EtOAc (200 mL). The organic layer was separated and the water layer was re-extracted with EtOAc (100 mL). The extracts were combined, washed with water and brine and evaporated in vacuo. The residue was purified by flash chromatography (50 g of silica gel) using EtOAc-Hexane (1:5 to 1:0) as an eluent, to give **6a** (8.8 g, 91%) as a colorless amorphous solid. ^1^H NMR: δ 9.90 (s, 1H), 8.24 (s, 1H), 8.14–8.08 (m, 1H), 7.60 (d, 1H, J = 8.0 Hz), 7.33–7.20 (m, 2H), 5.03 (t, 1H, J = 5.2 Hz), 4.30 (t, 2H, J = 5.3 Hz), 3.76 (q, 2H, J = 5.2 Hz). ^13^C NMR: δ 185.08, 141.97, 137.70, 125.14, 123.83, 122.84, 121.42, 117.41, 111.63, 60.04, 49.51. HRMS (EI) m/z [M + H]^+^ Calcd for C_11_H_12_NO_2_^+^ 190.0863; Found 190.0872.

1-(3-Hydroxypropyl)-1*H*-indole-3-carbaldehyde **(6b)**.

The same procedure as above was carried out using 3-(3-Formyl-1*H*-indol-1-yl)propyl acetate **(5b)** (6.0 g, 24 mmol), to give **6b** (4.5 g, 92%) as a colorless amorphous solid. ^1^H NMR: δ 9.89 (s, 1H), 8.26 (s, 1H), 8.10 (d, 1H, J = 7.5 Hz), 7.59 (d, 1H, J = 8.1 Hz), 7.35–7.21 (m, 2H), 4.73 (t, 1H, J = 5.0), 4.32 (t, 2H, J = 7.0 Hz), 3.39 (dd, 2H, J = 11.2, 5.9 Hz), 1.94 (p, 2H, J = 6.5 Hz). ^13^C NMR: δ 184.99, 141.29, 137.45, 125.11, 123.96, 122.90, 121.50, 117.48, 111.44, 57.99, 43.82, 32.79. HRMS (EI) m/z [M + H]^+^ Calcd for C_12_H_14_NO_2_^+^ 204.1019; Found 204.1030.

1-(4-Hydroxybutyl)-1*H*-indole-3-carbaldehyde **(6c)**.

The same procedure as above was carried out using 4-(3-formyl-1*H*-indol-1-yl)butyl acetate **5c** (5.0 g, 19 mmol), to give **6c** (3.7 g, 90%) as a colorless amorphous solid. ^1^H NMR: δ 9.89 (s, 1H), 8.29 (s, 1H), 8.10 (d, 1H, J = 7.6 Hz), 7.60 (d, 1H, J = 8.1 Hz), 7.33–7.21 (m, 2H), 4.47 (t, 1H, J = 5.1 Hz), 4.27 (t, 2H, J = 7.1 Hz), 1.90–1.76 (m, 2H), 1.47–1.33 (m, 2H). ^13^C NMR: δ 184.93, 141.11, 137.46, 125.14, 123.93, 122.86, 121.49, 117.49, 111.49, 60.64, 46.69, 29.94, 26.60. HRMS (EI) m/z [M + H]^+^ Calcd for C_13_H_16_NO_2_^+^ 218.1176; Found 218.1186.

1-(5-Hydroxypentyl)-1*H*-indole-3-carbaldehyde **(6d)**.

The same procedure as above was carried out using 5-(3-formyl-1*H*-indol-1-yl)pentyl acetate **5d** (5.0 g, 18 mmol), to give **6d** (3.9 g, 93%) as a colorless amorphous solid. ^1^H NMR: δ 9.88 (s, 1H), 8.29 (s, 1H), 8.13–8.06 (m, 1H), 7.59 (d, 1H, J = 8.2 Hz), 7.32–7.20 (m, 2H), 4.41 (br, 1H), 4.24 (t, 2H, J = 7.0 Hz), 3.34 (t, 2H, J = 6.4 Hz), 1.78 (p, 2H, J = 7.2 Hz), 1.41 (p, 2H, J = 6.7 Hz), 1.31–1.20 (m, 2H). ^13^C NMR (100 MHz, dmso) δ 185.00, 141.21, 137.46, 125.12, 123.96, 122.89, 121.51, 117.47, 111.49, 106.48, 60.90, 46.79, 32.36, 29.66, 23.18. HRMS (EI) m/z [M + H]^+^ Calcd for C_14_H_18_NO_2_^+^ 232.1332; Found 232.1338.

1-(6-Hydroxyhexyl)-1*H*-indole-3-carbaldehyde **(6e)**.

The same procedure as above was carried out using 6-(3-formyl-1*H*-indol-1-yl)hexyl acetate **5e** (6.0 g, 20 mmol), to give **6e** (4.8 g, 94%) as a colorless amorphous solid. ^1^H NMR: δ 9.88 (s, 1H), 8.29 (s, 1H), 8.09 (d, 1H, J = 7.7 Hz), 7.59 (d, 1H, J = 8.1 Hz), 7.33–7.19 (m, 2H), 4.38 (s, 1H), 4.24 (t, 2H J = 7.1 Hz), 3.33 (t, 2H, J = 6.3 Hz), 1.77 (p, 2H, J = 7.1 Hz), 1.41–1.19 (m, 6H). ^13^C NMR: δ 185.11, 140.95, 137.09, 125.33, 123.99, 122.77, 121.82, 117.40, 111.85, 61.02, 45.91, 32.65, 30.34, 26.75, 26.77. HRMS (EI) m/z [M + H]^+^ Calcd for C_15_H_20_NO_2_^+^ 246.1489; Found 246.1483.

*tris*(1-[2-Hydroxyethyl]-1*H*-indol-3-yl)methane **(7a)**.

To the solution of 2-indol-1-yl-ethanol **3a** (3.0 g, 18.7 mmol) in MeOH (100 mL), 1-(2-hydroxyethyl)-1*H*-indole-3-carbaldehyde **6a** (1.7 g, 9.1 mmol), AcOH (1 mL), and Dy(OTf)_3_ (10 mg, 16.4 μmol) were added. After refluxing for 12 h, the reaction mixture was cooled to RT. The resulting suspension was filtered, the precipitate washed with MeOH (2 × 50 mL) and Et_2_O (50 mL). The residue was dried in vacuo, to give **7a** (3.9 g, 89%) as a colorless amorphous solid.

^1^H NMR: δ 7.52–7.32 (m, 6H), 7.05 (t, 3H, J = 7.5 Hz), 7.01 (s, 3H), 6.88 (t, 3H, J = 7.3 Hz), 6.03 (s, 1H), 4.80 (t, 3H, J = 5.0 Hz), 4.11 (t, 6H, J = 5.1 Hz), 3.64 (d, 6H, J = 5.3 Hz). ^13^C NMR: δ 136.48, 127.25, 127.11, 120.59, 119.39, 118.0, 117.33, 109.77, 60.36, 48.12. HRMS (EI) m/z [M − H]+ Calcd for C_34_H_37_N_3_O_3_^+^ 493.2365; Found 492.2282.

*tris*(1-[3-Hydroxypropyl]-1*H*-indol-3-yl)methane **(7b)**.

The same procedure as above was carried out using 3-indol-1-yl-propan-1-ol **3b** (3.0 g, 17.1 mmol) and 1-(3-hydroxypropyl)-1*H*-indole-3-carbaldehyde **6b** (1.7 g, 8.5 mmol), to give **7b** (3.8 g, 85%) as a colorless amorphous solid. ^1^H NMR: δ 7.39 (t, 6H, J = 7.5 Hz), 7.05 (d, 3H, J = 7.9 Hz), 6.98 (s, 3H), 6.88 (d, 3H, J = 7.7 Hz), 6.03 (s, 1H), 4.52 (t, 3H, J = 5.0 Hz), 4.13 (t, 6H, J = 6.8 Hz), 3.42–3.23 (m, 6H), 1.89–1.67 (m, 6H). ^13^C NMR: δ 136.2, 127.05, 126.69, 120.71, 119.55, 117.99, 117.24, 109.63, 57.79, 42.2, 33.03. HRMS (EI) m/z [M − H] + Calcd for C_34_H_37_N_3_O_3_^+^ 535.2835; Found 534.2747. 

*tris*(1-[4-Hydroxybutyl]-1*H*-indol-3-yl)methane **(7c)**.

The same procedure as above was carried out using 4-indol-1-yl-butan-1-ol **3c** (3.0 g, 15.8 mmol) and 1-(4-Hydroxybutyl)-1*H*-indole-3-carbaldehyde **6c** (1.6 g, 7.8 mmol), to give **7c** (3.7 g, 82%) as a colorless amorphous solid. ^1^H NMR: δ 7.40 (d, J = 2.8 Hz, 3H), 7.38 (d, J = 3.3 Hz, 3H), 7.05 (t, J = 7.5 Hz, 3H), 6.96 (s, 3H), 6.86 (t, J = 7.4 Hz, 3H), 6.03 (s, 1H), 4.44 (t, J = 5.0 Hz, 3H), 4.07 (t, J = 6.8 Hz, 6H), 3.35 (q, J = 6.1 Hz, 6H), 1.90–1.48 (m, 6H), 1.48–1.11 (m, 6H). ^13^C NMR: δ 157.88, 144.03, 138.55, 124.44, 123.13, 120.55, 112.03, 59.90, 46.90, 29.14, 25.85. HRMS (EI) m/z [M − H] + Calcd for C_37_H_43_N_3_O_3_^+^ 577.3304; Found 576.3213.

*tris*(1-[5-Hydroxypentyl]-1*H*-indol-3-yl)methane **(7d)**.

The same procedure as above was carried out using 5-indol-1-yl-pentan-1-ol **3d** (3.0 g, 15.8 mmol) and 1-(5-hydroxypentyl)-1*H*-indole-3-carbaldehyde **5d** (1.6 g, 7.8 mmol), to give **7d** (4.2 g, 88%) as a colorless amorphous solid. ^1^H NMR: δ 7.38 (d, 6H, J = 5.3 Hz), 7.05 (t, 3H, J = 7.3 Hz), 6.95 (s, 3H), 6.86 (t, 3H, J = 7.3 Hz), 6.02 (s, 1H), 4.33 (t, 3H, J = 4.5 Hz), 4.06 (br, 6H), 1.65 (t, 6H, J = 6.8 Hz), 1.39 (t, 6H, J = 6.7 Hz), 1.37–0.95 (m, 6H). ^13^C NMR: δ 136.2, 127.05, 126.67, 120.68, 119.59, 117.91, 117.14, 109.64, 60.56, 45.24, 32.07, 29.69, 22.76. HRMS (EI) m/z [M] + Calcd for C_40_H_49_N_3_O_3_^+^ 619.3774; Found 618.3677.

*tris*(1-[6-Hydroxyhexyl]-1*H*-indol-3-yl)methane (**7e)**.

The same procedure as above was carried out using 6-indol-1-yl-hexan-1-ol **3e** (3.0 g, 13.8 mmol) and 1-(6-hydroxyhexyl)-1*H*-indole-3-carbaldehyde **6e** (1.7 g, 6.9 mmol), to give **7e** (3.7 g, 83%) as a colorless amorphous solid. ^1^H NMR: δ 7.38 (d, 6H, J = 8.6 Hz), 7.05 (t, 3H, J = 7.6 Hz), 6.93 (s, 3H), 6.85 (t, 3H, J = 7.7 Hz), 6.02 (s, 1H), 4.30 (t, 3H, J = 5.1 Hz), 4.06 (t, 6H, J = 6.7 Hz), 3.40–3.20 (m, 6H), 1.73–1.52 (m, 6H), 1.40–1.05 (m, 18H). ^13^C NMR (100 MHz, DMSO) δ = 136.18, 127.03, 126.67, 120.67, 119.56, 117.89, 117.09, 109.63, 60.54, 45.15, 32.43, 29.80, 26.07, 25.08. HRMS (EI) m/z [M]+ Calcd for C_43_H_55_N_3_O_3_^+^ 661.4243; Found 660.4160.

*tris*(1-(2-Hydroxyethyl)-1*H*-indol-3-yl)methylium chloride **(8a)**.

To the solution of *tris*(1-[2-hydroxyethyl]-1*H*-indol-3-yl)methane **7a** (2.0 g, 4.0 mmol) in THF (50 mL), DDQ (0.9 g, 4.0 mmol) was added. After stirring at rt for 1 h, the reaction mixture was quenched with conc. HCl (0.4 mL, 5 mmol) and evaporated in vacuo. The residue was purified by flash chromatography (100 g of silica gel) using CH_2_Cl_2_-MeOH (100:1 to 10:1) as an eluent to give **8a** (1.6 g, 78%) as a red amorphous solid. R_t_ = 3.39 min. ^1^H NMR: δ 8.35 (s, 3H), 7.84 (d, 3H, J = 8.3 Hz), 7.40 (t, 3H, J = 7.9 Hz), 7.15–7.05 (m, 6H), 4.92 (s, 3H), 4.51 (t, 6H, J = 5.2 Hz), 3.93 (t, 6H). ^13^C NMR: δ 144.8, 138.69, 126.74, 124.22, 122.98, 120.60, 117.42, 112.00, 59.12, 49.57. HRMS (EI) m/z [M]+ Calcd for C_31_H_30_N_3_O_3_^+^ 492.2282; Found 492.2270.

*tris*(1-(3-Hydroxypropyl)-1*H*-indol-3-yl)methylium chloride **(8b)**.

The same procedure as above was carried out using *tris*(1-[3-hydroxypropyl]-1*H*-indol-3-yl)methane **7b** (2.0 g, 3.7 mmol), to give **8b** (1.6 g, 73%) as a red amorphous solid. R_t_ = 3.55 min. ^1^H NMR: δ 8.38 (s, 3H), 7.83 (d, 3H, J = 8.3 Hz), 7.41 (t, 3H, J = 7.6 Hz), 7.13 (t, 3H, J = 7.6 Hz), 7.02 (s, 3H), 4.52 (t, 6H, J = 6.9 Hz), 4.46 (br, 3H), 3.56 (t, 6H, J = 5.9 Hz), 2.32–1.92 (m, 6H). ^13^C NMR: δ 157.77, 144.25, 138.49, 126.7, 124.35, 123.06, 120.49, 111.89, 57.47, 44.22, 31.89. HRMS (EI) m/z [M]+ Calcd for C_34_H_36_N_3_O_3_^+^ 534.2751; Found 534.2745.

*tris*(1-(4-Hydroxybutyl)-1*H*-indol-3-yl)methylium chloride **(8c)**.

The same procedure as above was carried out using *tris*(1-[4-hydroxybutyl]-1*H*-indol-3- yl)methane **7c** (2.0 g, 3.4 mmol), to give **8c** (1.7 g, 81%) as a red amorphous solid. R_t_ = 8.76 min. ^1^H NMR: δ 8.42 (s, 3H), 7.83 (d, J = 8.3 Hz, 3H), 7.40 (t, J = 7.6 Hz, 3H), 7.12 (t, J = 7.5 Hz, 3H), 6.99 (d, J = 7.4 Hz, 3H), 4.49 (t, J = 7.1 Hz, 6H), 4.37 (s, 3H), 3.49 (t, J = 6.2 Hz, 6H), 2.15–1.90 (m, 6H), 1.68–1.47 (m, 6H). ^13^C NMR: δ 157.88, 144.03, 138.55, 126.78, 124.44, 123.13, 120.55, 117.4, 112.03, 59.90, 46.90, 29.14, 25.85. HRMS (EI) m/z [M]+ Calcd for C_37_H_42_N_3_O_3_^+^ 576.3221; Found 576.3227.

*tris*(1-(5-Hydroxypentyl)-1*H*-indol-3-yl)methylium chloride **(8d)**.

The same procedure as above was carried out using *tris*(1-[5-hydroxypentyl]-1*H*-indol-3-yl)methane **7d** (2.0 g, 3.2 mmol), to give **8d** (1.8 g, 86%) as a red amorphous solid. R_t_ = 11.94 min. ^1^H NMR: δ 8.42 (s, 3H), 7.84 (d, 3H, J = 8.3 Hz), 7.41 (t, 3H, J = 7.6 Hz), 7.13 (t, 3H, J = 7.5 Hz), 7.01 (s, 3H), 4.46 (t, 6H, J = 7.0 Hz), 4.17 (s, 3H), 3.44 (br, 6H), 2.20–1.76 (m, 6H), 1.57–1.40 (m, 12H). ^13^C NMR: δ 138.47, 124.35, 123.06, 120.48, 111.94, 60.12, 46.93, 31.52, 28.73, 22.38. HRMS (EI) m/z [M]+ Calcd for C_40_H_48_N_3_O_3_^+^ 618.3690; Found 618.3682.

*tris*(1-(6-hydroxyhexyl)-1*H*-indol-3-yl)methylium chloride **(8e)**.

The same procedure as above was carried out using *tris*(1-[6-hydroxyhexyl]-1*H*-indol-3- yl)methane **7e** (2.0 g, 3.0 mmol), to give **8e** (1.7 g, 81%) as a red amorphous solid. R_t_ = 16.08 min. ^1^H NMR: δ 8.42 (s, 3H), 7.84 (d, J = 8.3 Hz, 3H), 7.41 (t, J = 7.7 Hz, 3H), 7.12 (t, J = 7.5 Hz, 3H), 7.00 (d, J = 7.1 Hz, 3H), 4.45 (t, J = 7.1 Hz, 6H), 4.09 (br, 3H), 3.41 (t, J = 6.1 Hz, 6H), 2.17–1.75 (m, 6H), 1.50–1.35 (m, 18H). ^13^C NMR: δ 157.82, 143.94, 138.49, 126.76, 124.36, 123.03, 120.44, 111.93, 60.24, 46.89, 31.87, 28.87, 25.61, 24.68. HRMS (EI) m/z [M]+ Calcd for C_43_H_54_N_3_O_3_^+^ 660.4160; Found 660.4150.

## 4. Conclusions

The first 5 representatives of *N*-(hydroxyalkyl) derivatives of *tris*(1*H*-indol-3-yl)methylium salts were synthesized and tested. Substances **8d** and **8e** showed high activity on Gram-positive bacteria, including resistant strains, and slightly less on Gram-negative ones. At the same time, the cytotoxicity of **8d** was 13 times lower than the antibacterial activity, which indicates the possible prospects for further search among this group of substances. Despite the fact that the exact target of these substances has not yet been established, it is known that the mechanism of their action is associated with a disruption of the membrane. Analysis of the structure-activity relationship showed an empirical dependence of the ratio of antibacterial/cytotoxic activity on the lipophilicity of the molecule. It is found that the best ratio is most likely achieved with LogP_ow_ close to 4. The possibility of theoretical calculation of LogP_ow_ for predicting the activity of new molecules using the Molinspiration package is shown.

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
