# Peer review of "N-(Hydroxyalkyl) Derivatives of tris(1H-indol-3-yl)methylium Salts as Promising Antibacterial Agents: Synthesis and Biological Evaluation"

_pharmaceuticals, 2020, doi:10.3390/ph13120469_

Round 1

Reviewer 1 Report

The authors described synthesis of novel N-hydroxyalkyl derivatives of tris(1H-indol-3-yl)methylium salts – analogues of turbomycin originally isolated as a product of Saccharomyces cerevisiae fermentation. Similar compounds have been reported by the same authors previously.  In my opinion the main point of this contribution pertains to the link between biological activity and lipophilicity/solubility of the synthesized analogues. Hence, authors conclude that the optimal logP parameter for this pharmacophore is c.a. 4. Unfortunately, experimental details regarding this study are poorly presented. Authors state that they used HPLC for logP determinations but experimental details are missing. Also in terms of theoretical determinations ‘Molinspiration package’ sounds rather cryptic. This part of manuscript certainly requires revision. In this regard literature references may be helpful. For example:

Int. J. Mol. Sci. 2019, 20, 5288; doi:10.3390/ijms20215288

In some cases English language style is lacking. A number of sentences require improvement.

Line 32

and possessed the ratio between antibacterial and cytotoxic activity more than 10 times.

Line 59

They had a relatively low toxicity to human cells, while having a high activity on resistant strains of Gram-positive bacteria [14] by disrupting the functioning of their membranes

Line 60

Analyzing the structure-activity relationship for these compounds, it was observed that the activity of these substances is closely related to their lipophilicity

Line 65

Thus, we made an assumption about the optimal

Line 78

different synthesis method was used

Line 90

Synthesis of all compounds

It would be easier to read Scheme 1 if compounds were described within the Scheme in the following way

2-8a: n= 1

2-8b: n = 2

….

Summing up,

I recommend this manuscript for publication on condition, however, that authors introduce suggested revisions.

Author Response

Dear reviewer! The authors are very grateful for your careful reading of the manuscript and comments that will help improve it!

all comments have been corrected:

Line 32 and possessed the ratio between antibacterial and cytotoxic activity more than 10 times.

Line 59 They had a relatively low toxicity to human cells, while having a high activity on resistant strains of Gram-positive bacteria [14] by disrupting the functioning of their membranes

Line 60 Analyzing the structure-activity relationship for these compounds, it was observed that the activity of these substances is closely related to their lipophilicity

Line 65 Thus, we made an assumption about the optimal

Line 78 different synthesis method was used

Line 90 Synthesis of all compounds

It would be easier to read Scheme 1 if compounds were described within the Scheme in the following way

2-8a: n= 1

2-8b: n = 2

As for : 

Unfortunately, experimental details regarding this study are poorly presented.

-we tried to expand the experimental in more detail.

Authors state that they used HPLC for logP determinations but experimental details are missing.

-All missing data was added. 

Also in terms of theoretical determinations ‘Molinspiration package’ sounds rather cryptic. This part of manuscript certainly requires revision.

- Revised and improved.

In this regard literature references may be helpful. For example:Int. J. Mol. Sci. 2019, 20, 5288; doi:10.3390/ijms20215288

-Thank you for reference! It was very helpful and added in references.

Reviewer 2 Report

    In this study, authors try to elucidate the effect of substitutions in tris(1H-indol-3-yl)methylium derivatives on their antibacterial activity and toxicity to human cells. Their antibacterial activity in vitro was performed on 12 bacterial strains (drug resistant strains from clinical or ATCC strains). The cytotoxic effect of the compounds was determined with human postnatal fibroblasts cells by MTT assay. Results indicated the most active compound revealed higher antibacterial activity on MRSA than the reference drug levofloxacin. Overall, the results appear sound and . However, part of the materials and methods are missing and some of the data should be discussed with literature reviews.

Major

  1. In the materials and methods, authors did not describe how the compounds were synthesized and what kind of chemicals they used. In addition, the antibacterial activity, cytotoxic activity, and the relationship between lipophility and biological activity were all missing. It is not appropriate to described this part in the results and discussions. Please conform and reorganize.
  2. Authors described the “Results and Discussion” in section 2. However, it is more likely that this part is material and method.
  3. In section 3. “Discussion”? Did you mean Conclusion?
  4. In addition, please discuss the antibacterial and cytotoxic activity with appropriate literatures.

Minor

  1. In Line 23-24, abstract, “Determination of antimicrobial activity was performed by double serial dilution method in liquid nutrient medium.” is not needed in abstract. Please delete it.
  2. In the Table 1. Antibacterial and Cytotoxic activity of 8 a-c, the IC50 should be IC50. Line 131, 138, 140, and so on.

Author Response

Dear reviewer! The authors are very grateful for your careful reading of the manuscript and comments that will help improve it!

all comments have been corrected:

Minor

  1. In Line 23-24, abstract, “Determination of antimicrobial activity was performed by double serial dilution method in liquid nutrient medium.” is not needed in abstract. Please delete it.
  2. In the Table 1. Antibacterial and Cytotoxic activity of 8 a-c, the IC50 should be IC50. Line 131, 138, 140, and so on.

As for:

In the materials and methods, authors did not describe how the compounds were synthesized and what kind of chemicals they used. In addition, the antibacterial activity, cytotoxic activity, and the relationship between lipophility and biological activity were all missing. It is not appropriate to described this part in the results and discussions. Please conform and reorganize.

-all done and reorganize

Authors described the “Results and Discussion” in section 2. However, it is more likely that this part is material and method.

-reorganize

In section 3. “Discussion”? Did you mean Conclusion?

-Discussion changed to Conclusion 

In addition, please discuss the antibacterial and cytotoxic activity with appropriate literatures.

-we tried to expand the discussion in more detail

Reviewer 3 Report

This article offers a new potential approach against 12 bacterial strains. The indole derivatives studied appear to be promising. The information produced and the methodology are well presented. However, the results must be better discussed. Although the number of molecules is not very high, a more detailed discussion of structure-activity relationships must be presented. In my opinion this work could be accepted but after major revision.

I am not an English native speaker and I do not feel qualified to correct all the mistakes. Only several language mistakes, some of them repeated along the text, must be reviewed.

I find some little errors to fix:

  • Line 34: space after comma
  • Line 63: than
  • Line 90, scheme 1: Better if the concentration of inorganic solution is indicated
  • Line 90: use letter in bold for compounds
  • Line 129: cytotoxic
  • Line 129: compounds 8a-e not a-c
  • Line 164: Figure 1.

Line 183: sentence not clear

Author Response

Dear reviewer! The authors are very grateful for your careful reading of the manuscript and comments that will help improve it!

all errors have been corrected:

  • Line 34: space after comma
  • Line 63: than
  • Line 90, scheme 1: Better if the concentration of inorganic solution is indicated
  • Line 90: use letter in bold for compounds
  • Line 129: cytotoxic
  • Line 129: compounds 8a-e not a-c
  • Line 164: Figure 1.

Line 183: sentence not clear

As for: However, the results must be better discussed. Although the number of molecules is not very high, a more detailed discussion of structure-activity relationships must be presented.

-we tried to expand the discussion in more detail

Round 2

Reviewer 2 Report

Dear Authors

    All comments have been corrected and the part of discussion has been described in more detail. I suggest this manuscript to be accepted in present form.

Reviewer 3 Report

given the corrections made, the manuscript is suitable for publication